# A Quantitative Method for the Composition of 7B05 Cast-Rolled Aluminum Alloys Based on Micro-Beam X-ray Fluorescence Spectroscopy and Its Application in Element Segregation of Recrystallization

**DOI:** 10.3390/ma16041605

**Published:** 2023-02-15

**Authors:** Caichang Dong, Dandan Sun, Dongling Li, Wanguo Yang, Haizhou Wang, Weihao Wan, Zun Yan

**Affiliations:** 1Qingdao NCS Testing & Corrosion Protection Technology Co., Ltd., Qingdao 266071, China; 2National Observation & Research Station on Materials Corrosion in Marine and Atmospheric Environment in Qingdao, Qingdao 266071, China; 3Beijing Key Laboratory of Metal Materials Characterization, Central Iron & Steel Research Institute, Beijing 100081, China

**Keywords:** X-ray fluorescence spectroscopy, 7B05 aluminum alloy, segregation index, recrystallization, microscopic segregation

## Abstract

Microscopic content segregation is among the important reasons for the anisotropy of mechanical properties in the cast-rolled sheets of the 7B05 aluminum alloy. It is of great significance to study the uniformity of aluminum alloys in terms of the microscopic composition and structure. In this study, a precise method for composition quantification based on micro-beam X-ray fluorescence spectroscopy is established by parameter optimization and a calibration coefficient. Furthermore, this method was applied for exploring and quantifying the relationship between recrystallization and deformation microstructures. The results show that the comprehensive measurement effects of all elements are the best when the X-ray tube voltage is 50 kV, the current is 150 μA, and the single-pixel scanning time is 100 ms. After verification, the sum of differences between the measured values and the standard values for all elements using the calibration coefficient is only 0.107%, which confirms the accuracy of the optimized quantitative method. Three types of segregation indexes in national standards were used to capture small differences, and finally ensure that the segregation degrees of elements are Ti > Fe > Cr > Cu > Mn > Zr > Zn > Al. The quantitative segregation results obtained by the spatial-mapping method show that the difference in the content of Al and Zn is approximately 0.2% between the recrystallization region and the deformation region, the difference in the content of Fe and Ti is 0.018% and 0.013%, the difference in the content of Cr, Cu and Zr is approximately 0.01%, and the difference in the content of Mn is not obvious, only 0.004%.

## 1. Introduction

Aluminum alloys are widely used in high-speed train components such as traction beams, corbel beams and buffer beams due to their characteristics of low density, corrosion resistance, weldability and thermal deformation [1,2]. There are obvious differences in the mechanical properties of hot-rolled thick plates of the 7B05 aluminum alloy along the thickness direction [3]. This anisotropy mainly results from the deformation and uneven temperature distribution during material processing and heat treatment [4]. The deformation in the surface layer is more serious than that in the center, because of the greater driving force caused by the interaction [5]. The center layer of the plate maintains at a high temperature for a long time, and the deformed fiber microstructure is easy to fuse with each other, forming coarse recrystallization grains [6]. As a result of the production process, the composition and microstructure distribution are asymmetrical, resulting in anisotropy in aspects of strength, plasticity, toughness and other properties [7,8,9,10].

The content segregation in different microstructure is an important reason for the difference in mechanical properties. How to mine and quantify the content relationship implicit in the microstructure is a key issue in characterization of the 7B05 material. Common analysis methods for component distribution, such as energy dispersive spectroscopy and electron probe microanalysis, are aimed at a precise test and used for characterization of single points and small regions [11,12,13]. Spark spectrometry and ICP are discontinuous-measurement techniques, so the global content distribution cannot be obtained [14,15]. The Original Position Statistical Distribution Analysis (OPA) and Laser-Induced Breakdown Spectroscopy (LIBS) can realize continuous distribution analysis in a large range, but there is a problem of surface damage [16]. In contrast, micro-beam X-ray fluorescence spectrometry has numerous advantages of high resolution in micro-area, fast analysis speed, no surface damage and sustainable testing [17,18]. By means of micro-beam X-ray fluorescence (μ-XRF), Li et al. [19] analyzed the surface composition distribution and depth profile of cobalt and superalloy composite powders mixed for different ball-milling times. Wang et al. [20] established a content analysis model for aluminum inclusions, the content of which is related to the number and intensity of spectral signals.

Due to the restriction of the production process, the uniformity in workpieces is difficult to guarantee, the abnormally growing recrystallization and element segregation make the plate’s side and intermediate performance present obvious differences. In this paper, a precise method for composition quantification in cast-rolled aluminum alloys based on micro-beam X-ray fluorescence spectroscopy is established to capture small differences between the recrystallization and deformation microstructures by way of parameter optimization and a calibration coefficient. Three kinds of segregation indexes in national standards were used to distinguish the microscopic segregation. Element segregation between recrystallization and deformation microstructures was explored and quantified through spatial mapping of two datasets [21,22,23]. By studying the distribution regularity, we reveal the evolution of microstructure in the cast and rolled aluminum alloy components, and then hope to guide their preparation process and improve material uniformity.

## 2. Materials and Method

### 2.1. Experimental Samples

The 7B05 aluminum alloy provided by China Railway Rolling Stock Corporation (CRRC) for commercial sleeper beams was used in the experiment. The elements of 7B05 were measured according to GB/T 20975-2020. Its chemical composition is shown in Table 1. This material was heat treated with artificial aging (T5) after hot rolling. Samples were cut from the rolled aluminum plate with a size of 10 mm × 10 mm × 15 mm (thickness), and the test surface remained perpendicular to the rolling direction (RD-ND). The sampling direction is shown in Figure 1.

### 2.2. Instruments and Conditions of Composition Test

The element composition distribution of the aluminum alloy section was analyzed by a micro-beam X-ray fluorescence spectrometer (M4 tornado, Bruker, Billerica, MA, USA). The structure and working principle of the instrument are shown in Figure 2. The detailed parameters are as follows: X-ray tube voltage 50 kV and current 150 μA, target material Rh, beam spot size 20 μm, beam spot collection interval 10 μm, scanning time per pixel 100 ms, and sample chamber vacuum 2000 Pa.

### 2.3. Optimization of Quantitative Methods

When the energy of the analytical line of the excited element in the sample is greater than the excitation energy of the spectral line of a coexisting element, the coexisting element will also strongly absorb the analytical line and be excited additionally. In the aluminum alloy element system, such interference from elements absorbing or enhancing effect is also unavoidable. In order to solve the spectral line interference in the process of element quantification, the calibration method for influence factor coefficient will be adopted in this paper. The quantitative method was optimized by using five series of bulk spectral samples with a similar composition to the 7B05 aluminum alloy. Through the linear relationship between the instrument-determined values and the certified values of standard samples of aluminum alloy series, the calibration coefficient (μ) is determined. The chemical composition of B4 series spectral standard samples used in this method is shown in Table 2.

### 2.4. Selection of the Segregation Index

After surface scanning of the aluminum alloy, approximately 1.5 million composition data of a single element were obtained within 150 square millimeters. How to count such a large amount of data, and then accurately characterize the distribution trend of a single element is a key problem. In this paper, three different statistical methods were chosen to quantitatively compare the segregation degree of elements. Firstly, the Original Statistical Distribution Analysis technology (GB/T24213-2009) was used to analyze the scanning data from micro-beam X-ray fluorescence spectroscopy. In this technology, S (statistical segregation degree) is the segregation degree of the content confidence interval [c_1_, c_2_] centered on the content median at 95% confidence level.
(1)$$S=\left(c_{2}-c_{1}\right)/2c_{0}$$

In Formula (1), c_0_ is the median value of the content, and the larger the statistical segregation degree S is, the more serious the element segregation is.

DS (the degree of segregation) is defined as the ratio of segregation to the overall average value, as shown in Formula (2).
(2)$$\mathrm{DS}=\frac{S_{E}}{x}=\frac{c-x}{x}$$

In Formula (2), c is the contents c_0.975_ and c_0.025_ corresponding to the confidence probabilities 97.5% and 2.5%, and x is the overall average value. When the confidence interval is 97.5%, DS is positive segregation, when the confidence interval is 2.5%, DS is negative segregation. The greater the absolute value of DS, the more severe the segregation.

Referring to GB/T37793-2019 quantitative analysis method for dendrite segregation of steel billets, arrange the element content in ascending order, the data with the same content occupying the space in sequence. The total number is N, and the cumulative frequency X is obtained by the sequence number of data divided by N. Equation (3) is the definition of SR_x_ (segregation ratio).
(3)$${\mathrm{SR}}_{x}=\frac{C_{\max}}{C_{\min}}$$

In Formula (3), SRx is the segregation ratio at cumulative frequency X. C_min_ is the arithmetic mean of the minimum content interval (0~X). C_max_ is the arithmetic mean of the maximum content interval (1 − X~1). According to GB/T37793-2019, the cumulative frequency X can be 0.05~0.2, and X is recommended to be 0.15 unless otherwise specified.

## 3. Experimental Results

### 3.1. Optimization of Measurement Parameters in Surface Distribution Mode

In order to obtain accurate signal intensity of each element and ensure the reliability of quantitative results, it is necessary to optimize the acquisition parameters. The optimized parameters include the general instrument’s tube voltage, tube current and single pixel acquisition time; the tube voltage and current ranges are 10~50 kV and 100~600 μA, respectively. Due to the large range span, this paper is expected to obtain the change trend of the indicators under different parameters through a set of orthogonal experiments. There are three types of indicators to screen the parameters: the peak-to-background ratio of all element, the relative standard deviation (rsd), and the relative content deviation (rcd). The scanning area was selected as 1 mm × 1 mm with relatively uniform distribution of elements. The experimental parameter settings are shown in Table 3. The scanning position is not moved during the acquisition process, so as to ensure that the data under all parameter conditions come from the same area.

The sum of the peak-to-back ratio of all elements under nine groups of conditions is shown in Figure 3. Due to the large net intensity of the matrix element, the peak-to-back ratio is usually greater than 1000, and the matrix fluctuation will cover the change in the peak-to-back ratio of other elements. Therefore, the sum of the peak-to-back ratios shown in Figure 3 does not include the matrix. As shown in Figure 3, as the tube power increases, the sum of the peak-to-back ratios tends to increase as a whole. When the current remains unchanged, the sum of the peak-to-back ratios is proportional to the voltage. The rsd represents the fluctuation of quantitative data, but the accuracy of the quantitative value cannot be assessed. Therefore, rcd (content deviation value = (quantitative result − actual result)/quantitative result) is used to assess the accuracy of quantification results. The value of rcd sum is consistent with that of rsd sum, both of which are first decreased and then increased, and the deviation is very large in the combination of small voltage, small current and large current.

It can be seen from Figure 3 that when the peak-to-back sum is very large or very small (the current is 600 μA or the voltage is 10 kV), the rsd and rcd of groups 1, 7, 8, and 9 tend to increase significantly; the fourth group is a combination of high current and low voltage, and the data collection time is 3-fold longer than that under normal situation. In order to avoid the occurrence of the above two situations, the selection range of the parameters is narrowed down to the 2nd, 3rd, 5th, and 6th groups. Compared with the data of groups 2 and 3, the increase in current does not significantly increase the peak-to-back ratio in group 5 and 6. That means the current’s contribution to the peak-to-back ratio is limited. The peak-to-back ratios of the three groups increase with the voltage, so the higher voltage can ensure the overall measurement effect of the material. Through the above analysis, the single-pixel acquisition time is not the only factor that determines the relative standard deviation, and it is more important to choose the appropriate tube current and voltage. In order to obtain higher peak-to-back ratio, this paper tends to choose a higher voltage. As can be seen from comprehensive evaluation of the peak-to-back ratio, rsd and rcd under 9 groups of conditions, the 3rd and 6th data groups ensure a larger peak-to-back ratio (220, 224), while rsd (1.62, 1.76) and rcd (3.01, 3.65) are small, which are ideal conditions.

The acquisition voltage optimized by the orthogonal experiment is 50 kV, and the optimal current is 100 or 350 μA. In terms of acquisition time, theoretically, with the increase in single-pixel acquisition time, the relative standard deviation will decrease. When a large-size cast-rolled sheet is measured, considering the limited signal counts of instrument, the single-pixel acquisition time of this material was set to 100 ms to avoid crash failure and ensure better stability. There is a large current span when setting the orthogonal experimental conditions. On the basis of determining the tube voltage (50 kV) and time (100 ms), in order to screen accurate current, seven groups of comparative experiments (50 μA~350 μA) were performed at an interval of 50 μA, with results shown in Figure 4.

Figure 4 shows the variation trend of the peak-to-back ratio of each element under different current conditions. With the increase in current, the peak-to-back ratio of two light elements, Al and Mg, decrease. Due to the high content of matrix elements and strong peak-to-back ratio signal, Al does not belong to the key elements to be investigated. Therefore, the sum of the peak-to-back ratios shown in Figure 4 does not include the matrix. Ti element shows a downward trend; Cr and Mn first decrease and then increase. Zn, Cu, Zr, and Fe as a whole show an increase and then a decrease. Figure 5 shows the rsd change trend of each element under seven groups of conditions. As the current increases, all elements show a gentle downward trend as a whole. Considering the peak-to-background ratio and rsd, the selectable current range of this material is 150~300 μA after optimization. The final measurement conditions are that the X-ray tube voltage is 50 kV, the current is 150 μA, and the single-pixel scanning time is 100 ms.

### 3.2. Optimization of Quantification Method

For optimization of the quantification method, five series of B4 aluminum alloy spectral standard samples were used to draw, and the composition range of standard samples includes the content of each element of the 7B05 alloy. The standard sample was collected in surface analysis under the measurement conditions described in Section 2.2, the collection area was 1 mm × 1 mm, and the linear relationships between the instrument-determined values and the certified values of standard samples of the aluminum alloy series are shown in Figure 6.

It can be seen from Figure 6 that the fitting slope of the matrix element is close to 1, meaning that there is basically no deviation between the standard value and the measured value. The slopes of Zr and Ti are close to 1, and the deviations between the standard values and the measured values do not exceed 3%. The deviations of Cr, Cu, and Mn elements are around 5%. The slope of Fe element is slightly higher, with a deviation approximately 12%; the deviation of Zn element is approximately 17%. The slope value of the Mg element fitting curve has a large deviation from 1, mainly because its atomic number is small and X-ray fluorescence intensity is low.

The value of calibration coefficient (μ) was determined by the slope of the fitting in Figure 6. In order to verify the effect of this method, a standard sample E323C (not used to establish this method) was selected from the B5 series aluminum alloy. The surface composition was collected under the same conditions, and μ was used for quantification. It can be seen from Table 4 that the measured content from the optimized quantitative method (μ-XRF) is basically consistent with the standard content (certified value). The sum of differences between the measured values and the standard values for all elements is only 0.107%, which confirms the accuracy and reliability of this method.

### 3.3. Application of Composition Analysis Method in the Aluminum Alloy

The quantitative analysis of each element in the range of 15 mm × 10 mm is carried out using the optimized quantitative method. In this method, the X direction is the thickness direction, and the number of component arrays is 1440 rows; the Y direction is the length direction, and the number of component arrays is 840 columns. The two-dimensional component distribution is shown in Figure 7. The results show that there are differences in the distribution of elements along the thickness direction of the rolled aluminum alloy plate—the distribution of elements in the surface layer is relatively uniform, and the distribution in the central layer presents a band-like segregation with a similar shape to that of the corroded structure.

In accordance with the GB/T24213-2009 on general rules of original position statistics, the content distribution analysis of each element at different positions obtained from the surface distribution was carried out. Statistical parameters such as the degree of segregation degree, statistical segregation degree, and segregation ratio of each element in the entire scanning area are obtained. For values of S and SRx, refer to Equations (1) and (3). Values of DS Positive and DS Negative correspond to confidence intervals of 97.5% and 2.5%, respectively, referring to Equation (2). The results are shown in Table 5.

Three different statistical methods are chosen to quantitatively compare the segregation degree of elements: S, SR_x_, DS Positive and DS Negative, as shown in Table 5. This table indicates S: Ti > Fe > Cr > Mn > Cu > Zr > Zn > Al; SR_x_: Ti > Fe > Cr > Cu > Mn > Zr > Zn > Al; DS Positive: Ti > Fe > Mn > Zr > Cr > Cu > Zn > Al; DS Negative: Ti > Fe > Cr > Cu > Mn > Zr > Zn > Al. The above four segregation indicators show that the segregation of Ti is the most serious, followed by Fe, and the distributions of Al, Zn, and Zr are the most uniform. The segregation degrees of Cr, Cu and Mn are similar, the S indicates that three elements are difficult to separate (0.204, 0.201, 0.203), and the SR_x_ and DS Negative indicates that three elements are relatively different. In summary, the results of four segregation evaluation indexes are basically consistent, which can be used to characterize small segregation difference in a complementary manner; the segregation degree of element in 7B05 is Ti > Fe > Cr > Cu > Mn > Zr > Zn > Al.

The frequency distribution diagram of element content is shown in Figure 8. The red curve is the normal distribution fitted by the frequency distribution map of each element, and the sparseness of the map is related to the total categories of content values. It can be seen from the figure that the frequency distribution basically conforms to the normal distribution, and the distribution of Al, Fe and Mn has a larger deviation from the normal distribution curve. Al has a tailing effect at low content, the total number of low content points higher than that of high content points. The number of Fe and Mn with high content points is more than that of lower content points, which may be related to the point-like segregation generated by the second phase in the cross section. The normal distribution curve of Ti has the flattest amplitude, indicating that the data distribution of Ti is more discrete from the average, and the data fluctuation is the largest. It is consistent with the calculated results of statistical segregation degree.

Furthermore, to explore the variation law of the content along the thickness direction in the aluminum alloy rolled sheet, the scatter diagram and fitting diagram of the average content are shown in Figure 9. As can be seen from Section 3.3, the mapping data in the section consist of 1440 columns and 840 rows. The average data of 840 rows corresponding to each column were extracted, and a total of 1440 data points were obtained along the thickness direction. Each point represents the average value of 840 rows and 1 column, which means each point represents the average content of 840 μm × 10 μm microdomains. The figure reveals that from the center of the rolled sheet to the left and right surfaces, the distribution of elements is asymmetric, and the overall content fluctuates greatly, but the element fluctuation is relatively small in the range of 5–1 mm. There is some connection in the distribution of elements. The regularities of Al and Zn elements show opposite trends, and the changing trends of Al and Cr, Zn and Fe, Cu and Ti are similar.

### 3.4. Refining Analysis of Component Distribution Results

Application results in Section 3.3 show that elements distribution is related to the morphology after corrosion. In order to further explore the regularity of distribution between elements and microstructure, this paper selected a 1 mm × 1 mm micro-area to mine the correlation. The relationship between the average content and position of each element is shown in Figure 10. The abscissa is the position coordinate, and ordinate is the average content distribution of each element. It can be seen from the figure that Al, Cr, Ti, and Zr have similar distribution laws, and Zn, Cu, Fe, and Mn have similar distribution laws. The black curve is the average gray value curve corresponding to the fluorescence spectrum. This figure also indicates that element content may be a correlation between the content distribution and the gray value of the optical image. The specific correlation will be further verified by the single-point scanning results.

In this paper, 60 positions with gray value differences were selected, respectively, and single-point test was carried out on them. The average content was obtained after three measurements at each position. These test results were divided into categories A and B according to gray value. Class A represents a low gray area, and Class B represents relatively high grayscale value. As shown in Table 6, Al, Cr and Ti are high in the position with low gray, and Zn, Fe, Cu, Mn elements have low content in the position with low gray.

Since the content and grayscale data are collected from two different kinds of scale and type, it is difficult to use point-to-point to explore the correlation between the two. This paper will explore rules between the two datasets by the inter-micro-area spatial-mapping method, and quantitatively express the degree of correlation between the two. The component data and grayscale data set in the range of 1 mm × 1 mm were divided into 16 regions, and the data in same region showed a one-to-one correlation, as shown in Figure 11.

It can be seen from Figure 11 that regions with high average gray values have high contents of Al, Cr, Ti, which indicates that these elements exist in the form of microscopic positive segregation. The areas with high average gray values have low Zn, Cu, Mn, Fe contents, indicating that these elements exist in the form of microscopic negative segregation. As is conveyed by Figure 7a, the white region with high gray value is generated from large recrystallized grains, which are resistant to corrosion due to their low internal crystal defects and dislocation density. The gray-black region has fine grains and is composed of deformed sub-grains. Due to crystal defects and dislocation density, a large number of sub-grain boundaries are formed, which are easily corroded, so that the structure appears black [3]. According to the above research results, Figure 11 is divided into two categories to calculate the element content of recrystallization and deformation microstructures. Calculation results are shown in Table 7. The difference in the content of Al and Zn is approximately 0.2%, the difference in the content of Fe and Ti is 0.018% and 0.013%, the difference in the content of Cr, Cu and Zr is approximately 0.01%. The difference in the content of Mn is not obvious, only 0.03%.

Figure 12a shows the microstructure morphology of 7B05-T5 after corrosion. EDS was used to verify the above fluorescence data. The entire field of view was collected in a mapping mode, and there was no difference between the deformation microstructure and the recrystallization microstructure even if the acquisition time was increased. Therefore, small range of tests were used for verification experiments. Quadrangle 1, 2, and 3 represent the recrystallization regions, where the grains are larger and are resistant to corrosion due to the low density of internal crystal defects and dislocations. Quadrangle 4, 5 and 6 represent the deformation regions, where the grains are fine and prone to corrosion, and the energy spectrum results is shown in Figure 12b. The differences in the contents between two regions is shown in Table 8. It can be seen from Table 8 that the trend of two groups is basically consistent with the fluorescence data, indicating that Al, Ti and Zr have high content in the recrystallization region. This also proves the reliability of the fluorescence data. Due to the limited of measured number and the small content of Zr and Ti in 7B05-T5, values measured by EDS are higher than actual values. At the same time, the verification test also suggests the existence of a large number of precipitated phases at the grain boundaries of the deformation region. The main alloying elements of the precipitated phase are Zn, Cu, Mn and Fe, which also explains the formation of a band-like distribution in fluorescence data.

## 4. Conclusions

(1)Considering the optimal measurement effect of all elements, the final measurement conditions are determined as follows: the X-ray tube voltage is 50 kV, the current is 150 μA, and the single-pixel scanning time is 100 ms.(2)Using the calibration coefficient and after verification with the E323C standard sample, the sum of differences between the measured values and the standard values for all elements is only 0.107%.(3)The segregation degree of element in 7B05 is Ti > Fe > Cr > Cu > Mn > Zr > Zn > Al; When the difference in element segregation degree is not large enough to distinguish, the three complementary indexes can be used to accurately characterize element segregation.(4)The distribution of elements presents a band-like segregation, which is similar to the microstructure morphology. Al, Cr, Ti, and Zr exist in the form of microscopic positive segregation, and Zn, Cu, Mn, and Fe exist in the form of microscopic negative segregation in the recrystallization region. The difference in the content of Al and Zn is approximately 0.2% between the recrystallization region and the deformation region, the difference in the content of Fe and Ti is 0.018% and 0.013%, the difference in the content of Cr, Cu and Zr is approximately 0.01%, and the difference in the content of Mn is not obvious, only 0.004%. Element segregation may be related to the distribution of the precipitated phase at grain boundaries.

## Figures and Tables

**Figure 1 materials-16-01605-f001:**
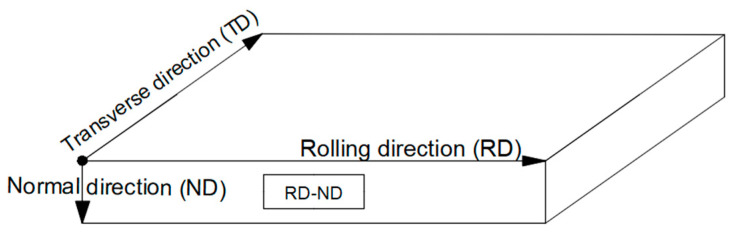
Schematic diagram of sampling location in a cast-rolled sheet.

**Figure 2 materials-16-01605-f002:**
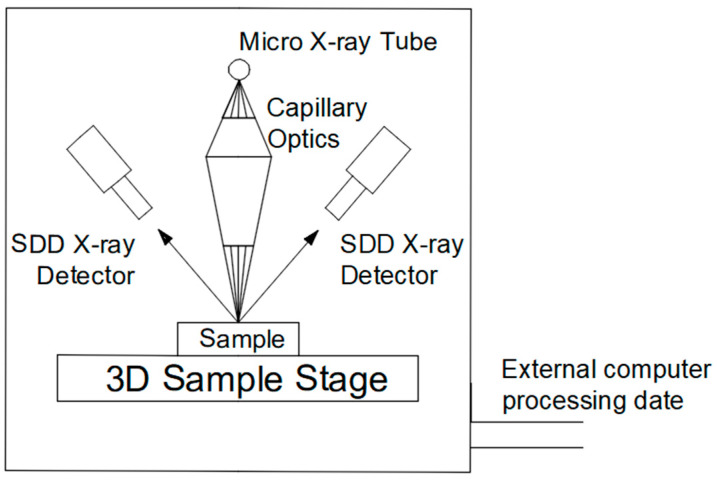
Schematic diagram of the structure of the micro-beam X-ray fluorescence spectrometer.

**Figure 3 materials-16-01605-f003:**
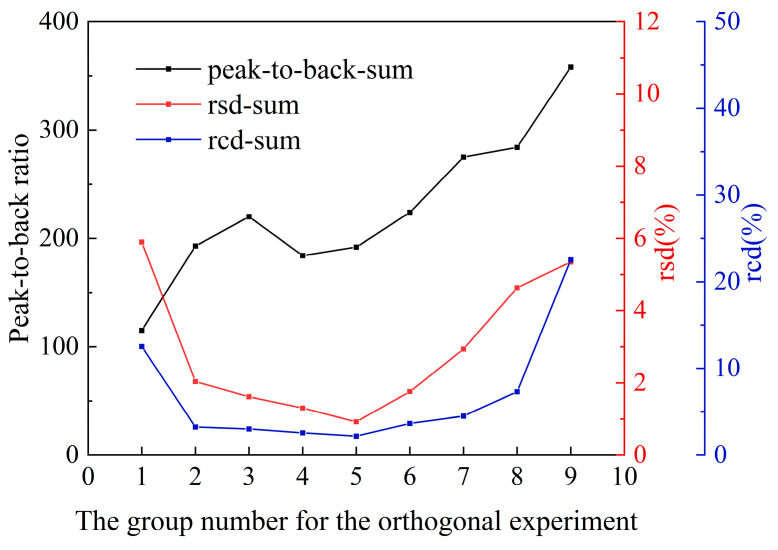
The corresponding peak-to-back ratio, the relative standard deviation and the relative content deviation of orthogonal test.

**Figure 4 materials-16-01605-f004:**
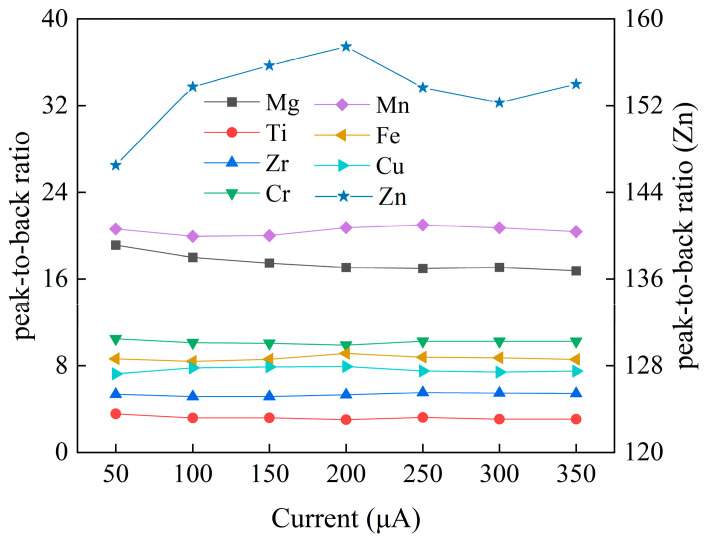
Peak-to-back ratio measured by X-ray tube at different currents.

**Figure 5 materials-16-01605-f005:**
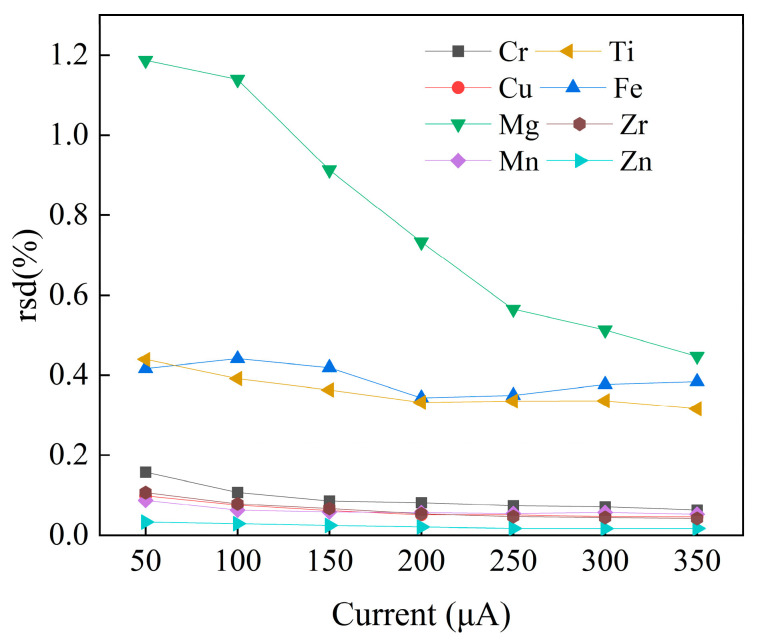
Standard deviation under different currents measured by X-ray tubes.

**Figure 6 materials-16-01605-f006:**
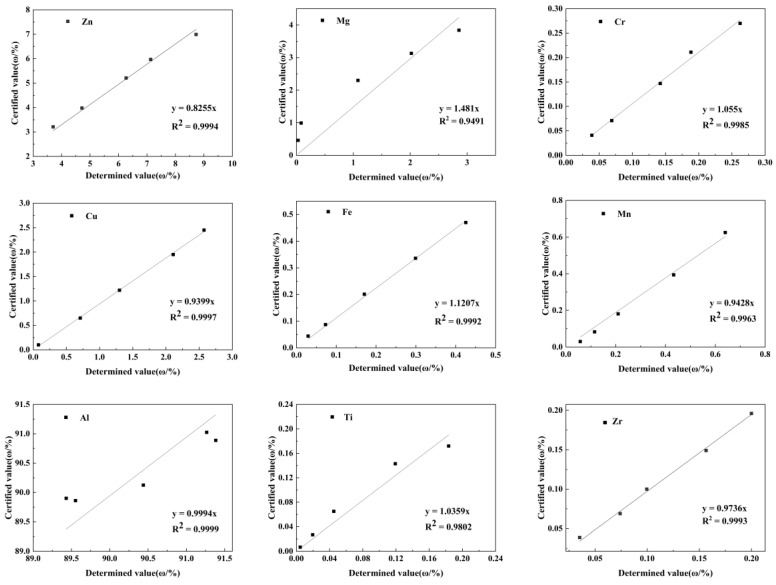
Linear fitting diagram of the μ-XRF measured value and certified value of the aluminum alloy standard sample.

**Figure 7 materials-16-01605-f007:**
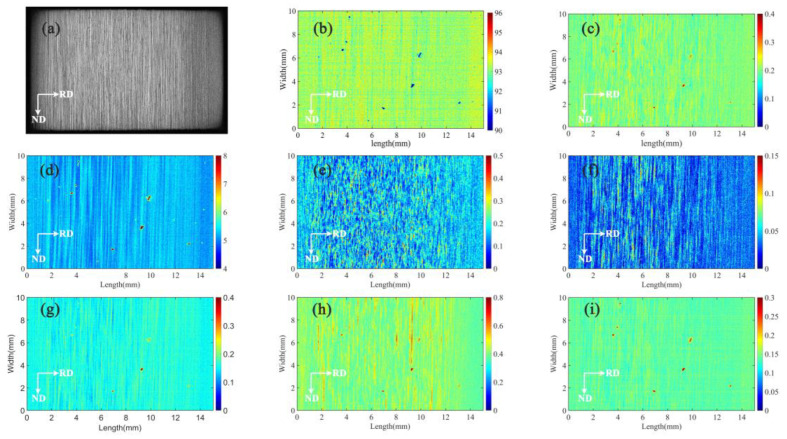
Two-dimensional distribution map of element content in 7B05 aluminum alloy; (**a**) metallographic structure after corrosion, (**b**) Al, (**c**) Cr, (**d**) Zn, (**e**) Fe, (**f**) Ti, (**g**) Cu, (**h**) Mn, and (**i**) Zr.

**Figure 8 materials-16-01605-f008:**
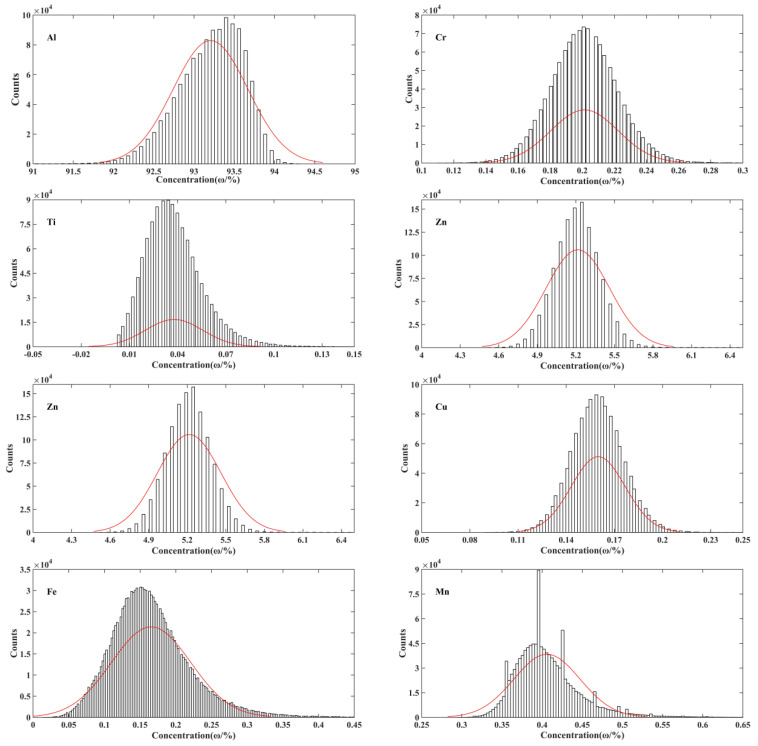
The frequency distribution histogram of element content in the 7B05 aluminum alloy.

**Figure 9 materials-16-01605-f009:**
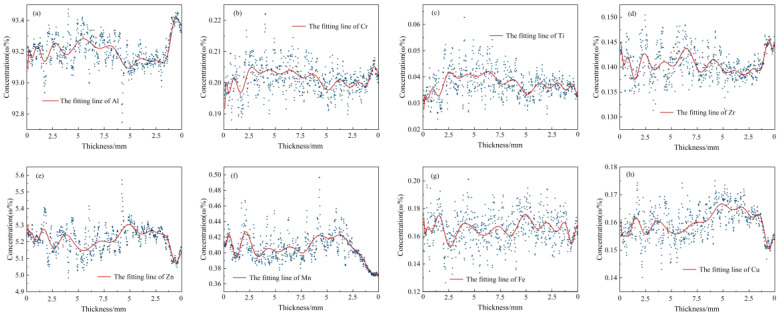
The average content line distribution along the thickness in the 7B05 aluminum alloy rolling plate; (**a**) Al, (**b**) Cr, (**c**) Ti, (**d**) Zr, (**e**) Zn, (**f**) Mn, (**g**) Fe, (**h**) Cu.

**Figure 10 materials-16-01605-f010:**
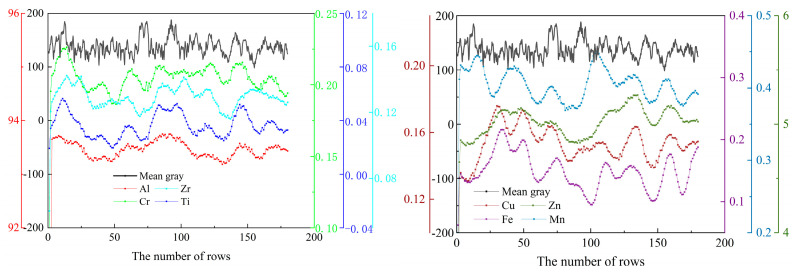
The average content line distribution in micro-region.

**Figure 11 materials-16-01605-f011:**
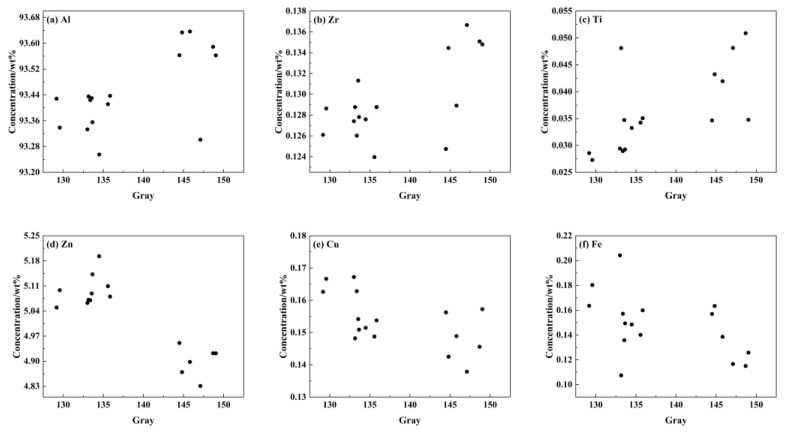
Quantitative relationship diagram between content and microstructure in the 7B05 aluminum alloy. (**a**) Al, (**b**) Zr, (**c**) Ti, (**d**) Zn, (**e**) Cu, (**f**) Fe.

**Figure 12 materials-16-01605-f012:**
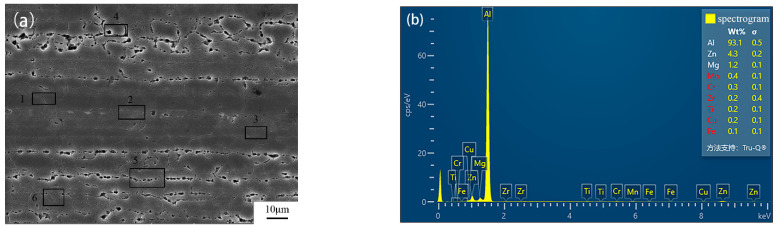
Microstructure and energy spectrum of 7B05-T5. (**a**)microstructure and (**b**) energy spectrum of the deformation region.

**Table 1 materials-16-01605-t001:** Chemical composition of the 7B05 aluminum alloys (wt%).

Material	Zn	Mg	Cu	Fe	Si	Mn	Cr	Zr	Ti
T5-15	4.23	1.09	0.16	0.17	0.058	0.37	0.22	0.11	0.048

**Table 2 materials-16-01605-t002:** The chemical composition of B4 series spectral standard samples (wt%).

Number	Zr	Zn	Ti	Si	Ni	Mn	Mg	Fe	Cu	Cr	Al
GSB-1	0.149	3.21	0.0068	0.043	0.0087	0.03	3.13	0.044	2.45	0.041	90.8875
GSB-2	0.1	3.98	0.027	0.085	0.052	0.083	3.84	0.087	0.65	0.071	91.025
GSB-3	0.039	5.21	0.143	0.206	0.093	0.181	2.3	0.336	1.22	0.147	90.125
GSB-4	0.196	5.97	0.065	0.42	0.175	0.394	0.46	0.201	1.95	0.27	89.899
GSB-5	0.069	6.99	0.172	0.49	0.022	0.625	0.99	0.47	0.101	0.211	89.86

**Table 3 materials-16-01605-t003:** Orthogonal experiment table of instrument parameter screening.

Number	Current (μA)	Voltage (kV)	Time (ms)	Peak-to-Back Sum	Rsd Sum (%)	Rcd Sum (%)
1	100	10	50	115	5.90	12.52
2	100	30	100	193	2.04	3.24
3	100	50	150	220	1.62	3.01
4	350	10	100	184	1.30	2.56
5	350	30	150	192	0.92	2.18
6	350	50	50	224	1.76	3.65
7	600	10	150	275	2.94	4.51
8	600	30	50	284	4.63	7.31
9	600	50	100	358	5.35	22.52

**Table 4 materials-16-01605-t004:** The verification results of calibration method for the influence factor coefficient.

Method	Zn	Cu	Mn	Cr	Zr	Fe	Ti	Al
μ	0.826	0.940	0.943	1.055	0.974	1.121	1.036	0.999
Before calibration	6.044	1.468	0.344	0.116	0.120	0.363	0.028	90.381
After calibration	4.992	1.380	0.324	0.122	0.117	0.407	0.029	90.327
Certified value	4.990	1.330	0.316	0.133	0.112	0.386	0.035	90.323
Content difference	0.002	0.05	0.008	0.011	0.005	0.021	0.006	0.004

**Table 5 materials-16-01605-t005:** Results of statistical distribution analysis of element content.

Index/Element	Al	Cr	Cu	Fe	Mn	Ti	Zn	Zr
Average	93.12 ± 0.188	0.21 ± 0.003	0.15 ± 0.004	0.19 ± 0.014	0.39 ± 0.013	0.04 ± 0.004	4.31 ± 0.146	0.14 ± 0.005
Maximum (97.5%)	93.77	0.25	0.18	0.34	0.48	0.08	4.59	0.17
Minimum (2.5%)	92.24	0.17	0.12	0.09	0.33	0.01	4.03	0.12
S	0.008	0.204	0.201	0.692	0.203	0.946	0.0653	0.175
DS Positive	0.007	0.200	0.188	0.765	0.244	1.000	0.065	0.214
DS Negative	−0.009	−0.200	−0.188	−0.529	−0.146	−0.750	−0.065	−0.143
SR_x_	1.014	1.38	1.374	2.841	1.344	5.182	1.116	1.317

**Table 6 materials-16-01605-t006:** Average content results from single point scan (wt%).

Position	Al	Ti	Cr	Mn	Fe	Cu	Zn	Zr
A-content	92.466±	0.042±	0.214±	0.417±	0.191±	0.174±	5.067±	0.140±
A-standard deviation	0.216	0.0195	0.0163	0.0234	0.0733	0.0130	0.139	0.0061
B-content	92.589±	0.046±	0.218±	0.411±	0.173±	0.172±	5.010±	0.138±
B-standard deviation	0.184	0.019	0.016	0.0340	0.0837	0.0120	0.0835	0.0044

**Table 7 materials-16-01605-t007:** The differences in the contents between recrystallization and deformation microstructures (wt%).

Region	Al	Cr	Cu	Fe	Mn	Ti	Zn	Zr
Deformation	93.292 ± 0.058	0.211 ± 0.003	0.148 ± 0.007	0.173 ± 0.013	0.378 ± 0.005	0.032 ± 0.003	4.210 ± 0.041	0.125 ± 0.001
Recrystallization	93.503 ± 0.033	0.220 ± 0.006	0.139 ± 0.007	0.152 ± 0.015	0.374 ± 0.009	0.046 ± 0.005	4.047 ± 0.039	0.131 ± 0.002
Content difference	0.212	0.009	0.008	0.020	0.004	0.013	0.164	0.007

**Table 8 materials-16-01605-t008:** EDS content differences between recrystallization and deformation microstructures (wt%).

Region	Al	Cr	Cu	Fe	Mn	Ti	Zn	Zr
Deformation	93.108 ± 0.141	0.277 ± 0.0208	0.185 ± 0.047	0.311 ± 0.658	0.348 ± 0.0314	0.126 ± 0.133	4.298 ± 0.878	0.221 ± 0.047
Recrystallization	93.400 ± 0.084	0.267 ± 0.016	0.167 ± 0.013	0.100 ± 0.001	0.333 ± 0.0310	0.133 ± 0.040	4.094 ± 0.426	0.235 ± 0.011
Content difference	0.292	0.01	0.018	0.211	0.014	0.007	0.204	0.013

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
