# Peer review of "A Quantitative Method for the Composition of 7B05 Cast-Rolled Aluminum Alloys Based on Micro-Beam X-ray Fluorescence Spectroscopy and Its Application in Element Segregation of Recrystallization"

_materials, 2023, doi:10.3390/ma16041605_

Round 1
Reviewer 1 Report
Please, see attached file

Author Response
In the manuscript entitled “A Quantitative Method for the Composition in 7B05 Cast-Rolled Aluminum Alloys Based on Micro Beam X-Ray Fluorescence Spectroscopy and Its Application in Element Segregation of Recrystallization” authors develop a quantitative method for determining the real composition of alloys based on micro beam X-ray fluorescence spectroscopy. This method was evaluated in a 7BO5 Cast-Rolled aluminum alloy. The results are new, interesting and important for possible applications, making these results a guide for the preparation process and a way to improve uniformity and homogeneity in this kind of systems. However, the following issues have to be addressed before considering the paper for publication:
- Typo in kV (First in line 50, but it is repeated in all the manuscript)
All “kv” has been modified to “kV” in the article.
- Table 1. Please, indicate if the chemical composition collected in it has been reported by the supplier of it has been measured. Please, include the error.
The chemical composition was measured by GB/T 20975-2020. The data was provided by the third-party testing company, and the specific error value was not calculated.
- Equation 2. Please, clarify the meaning of this equation.
When the confidence interval is 97.5%, DS is positive segregation, when the confidence interval is 2.5%, DS is negative segregation. A clear description has been added to the article.
- Equation 3. Please, justify the selection of X=0.15.
According to GB/T37793-2019, the cumulative frequency X can be 0.05~0.2, if there are no special provisions, X is recommended to be 0.15.
- Regarding Table 2. Please, justify the variation of different parameters at the same time. It is not easy obtain conclusions if two or more parameters are modified. Please, include the corresponding units and errors.
Corresponding explanation has been added in manuscript as follow: Compared with the data of groups 2, 3, the increase of current did not significantly increase the peak-to-back ratio in group 5, 6. That means current's contribution to the peak-to-back ratio is limited. The peak-to-back ratio of each 3 groups increased with the increase of voltage, so the higher voltage can ensure the overall measurement effect of the material (lines 175-179).
Corresponding units and errors have been added in Table2
- The conclusions in lines 162-166 about the optimal conditions of the measurements are unclear.
The conclusions in lines 162-166 about the optimal conditions of the measurements have been modified as follow: Compared with the data of groups 2, 3, the increase of current did not significantly increase the peak-to-back ratio in group 5, 6. That means current's contribution to the peak-to-back ratio is limited. The peak-to-back ratio of each 3 groups increased with the increase of voltage, so the higher voltage can ensure the overall measurement effect of the material. Through the above analysis, the single-pixel acquisition time is not the only factor that determines the relative standard deviation, and it is more important to choose the appropriate tube current and voltage. In order to obtain higher peak-to-back ratio, this paper tends to choose a higher voltage. Comprehensive evaluation of the peak-to-back ratio, rsd and rcd under 9 groups of conditions, the 3th and 6th data can ensure a larger peak-to-back ratio(220,224), while rsd(1.62,1.76) and rcd(3.01,3.65) are small, which is an ideal condition.
- The characteristics of the B4 aluminum alloy series employed to optimize the method should be indicated in the experimental section, which their corresponding composition.
The characteristics of the B4 aluminum alloy series employed to optimize the method have been indicated in 2.3 section (Optimization of quantitative methods).
- Figure 6. I don’t understand how it is possible that the best fitting corresponds to the case of the Al, when it is clear the dispersion of the points.
After verification, we found that the error between the value calculated by the fitting curve and the actual value is within the range of 2.5%-5.8%, which is fitting corresponds to the case of the Al. As the fluctuation between data is much smaller than the overall base, the coordinate spacing is set small, resulting in the fitting image looks discrete.
- Table 3. It would be interesting to include the obtained values before calibration was performed in order to compare the values.
The obtained values before calibration ware performed in Table 4.
- Please, explain the variation in the area of the sample in the different section of the manuscript.
When studying the distribution law of materials, the section area of 10 mm×15 mm is used in 3.3 section, because the larger area can more truly reflect the distribution state of elements. When exploring the relationship between the distribution of elements and the structure after corrosion, the area of 1 mm×1 mm is selected considering two aspects. First, the data of 1 mm×1 mm is close to 1000 points, and the amount of data is relatively less. The segmentation error for two data set is small, which is conducive to the precise quantification of the microdomain correspondence between data sets. Another reason is that microstructure distribution is not uniform in the cross section. The surface is equiaxed recrystallization, the central layer is deformed microstructure. Banded segregation occurs mainly in the recrystallization of the central layer. Therefore, it is necessary to select the central layer region for quantitative relationship exploration.
- Table 4. Please, include the estimated erros.
The estimated erros have been added in Table 4
- Figure 8. Could you explain the distribution fittings? Ti and Fe fittings don’t have physical meaning, with some contribution for values <0.
When large numbers of data are counted, the fitting tends to be normally distributed. The dispersion of elements can be judged by the shape of the normal distribution. Values less than 0 have no physical meaning and it is only the result of mathematical model fitting.
- Please, explain the concept microdomains (line 271)
The concept about microdomains in 271 have been explained. As can be seen from Section 3.3, the mapping data in the section consists of 1440 columns and 840 rows. The average data of 840 rows corresponding to each column was extracted, and a total of 1440 data points were obtained along the thickness direction. Each point represents the average value of 840 rows and a column, which means each point represents the average content of 840 μm×10 μm microdomains.
- It would be interesting to check the quantities obtained using the developed method with common analysis methods for component distribution, such as EDS.
This is a very enlightening suggestion. Related EDS experiment was not implemented due to time constraints. The two methods will be compared in the future according to this proposal.
- Table 5. Please, include errors and if the content refers to at.% or wt.%
“wt.%” has been added Table 5.
Standard deviation means error, and it has existed in Table 5
- It is unclear the meaning of x axis in Figure 11.
The meaning of x axis in Figure 11 is grayscale. As is conveyed by Fig.7(a), The gray scale data after corrosion is similar to composition distribution. Therefore, the gray scale value in the micro-area is taken as the horizontal coordinate and the content as the vertical coordinate to explore the relationship between them.
- Table 6. Idem Table 5
The estimated erros have been added in Table 6
- Please, reformulate the conclusion section.
The conclusion has been reformulated.

Reviewer 2 Report
The main advantage of this article is its completeness and explanation of all aspects, starting with the development and validation of a measurement method, ending with its practical application. I have no major objections to the content of this paper. However, in order to improve its quality and avoid misunderstandings, I propose a number of changes, which I have included in the comments below.
Overall remarks:
1) Editorially, the whole text needs to be corrected in some parts in detail, e.g. voltage should be in “kV”, not “kv”; there are sometimes wrong punctuation marks; RSD sometimes is written by capital letters and in other places by lowercase “rms”; and others. Moreover, wrong Tense was used in some places e.g. line 172.
2) Some sentences are very long (4 lines of text) and it cause, that finally are difficult to understand the main meaning of the sentence
Detailed remarks:
#1 line 20, 128 – correct “kv” into “kV”
#2 line 21 – phrase “the content deviation of all elements” is difficult to understand. If it is a difference between the sum of all measured contents of the elements and the 100% or if the summary of all individual deviations? It must be clearly described. (see also remark #17)
#3 lines 25-28 – all percentages should be in the same level of accuracy 0.01% or 0.001% (see also remark #24)
#4 lines 29-30 – delete “Optimization parameter” and “Calibration coefficient” – they are too general;
#5 lines 36-37 – correct the phrase “on the same plane and different planes along the thickness direction” to more understandable
#6 line 44 – what is it “plastic shape” ?
#7 lines 48-50 – the sentence “Common analysis methods for component distribution, such as scanning electron microscope, energy spectrum analysis and electron probe microanalysis, are only aimed at micro area test, with slow analysis speed and low quantitative sensitivity” is not true. Firstly, SEM is not the method of chemical analyse (but electron beam of SEM is used as a source of high energy electrons). What is mean “slow speed” – if 20 s is slow or fast? What is mean “low quantitative sensitivity” and if 0,1 % is low? These comments are too vague and can be compared to another method rather than as imperative statements.
#8 line 81 – data placed in table were measured, from standard or from otherwise? This information must be provided. (reference)
#9 line 88 – what does mean “target material was Rh”? Especially, what does mean “Rh”
#10 line 90 – the pressure in chamber should be in SI unit [Pa].
#11 lines 111-113 – there is not explained, what does mean “positive” and “negative” segregation.
#12 line 122 – why X was selected as 0.15?
#13 Table 2 – lack of units; not “Electricity” but “Current”
#14 line 128 – the current and voltage cannot be “0” – in table 2 are given minimal values
#15 figure 3 - the horizontal axis and vertical “blue” are not described
#16 figures 4 and 5 should be modified – at the horizontal axis are the values of current; lack of the Al signal, described in text
#17 lines 207-220 – there must be clearly defined if the % deviation is the deviation of the slope or it is deviation of the measured content of elements from standard value. Both types of deviations are used in text, but e.g. for (Zr, Ti) the deviation concerns rather slope than “measured value” -> see next sentence, where the deviation concerns the elements. You must avoid misunderstandings, like this. Moreover, there must be placed information in which way the value of “the content deviation of all elements” (0.107%) was determined (see. remark #2)
#18 table 3 – there is lack of Mg. Moreover, the sum of elements, both certified (!) and measured is not 100% (app. 97,6%). Why? Place in table additional line with percentage deviation, because you comment it in line 220.
#19 lines 226-227 – there should be the reference to Fig. 1 and the directions marked over there (ND, RD, TD)
#20 concerns the data placed in Table 4 - How were determined the values of S, DS positive, DS negative and SRx, referring to eq. (1), (2) and (3). In the Table 4 are not the symbols like used in the equations. There must be explanation or have to be written some additional formulas.
#21 lines 257-258 and figure 8 – it is rather the “density distribution” or “count distribution” than “frequency distribution”; please check it, because “frequency” is connected rather with wavy phenomena. The distributions model of Fe, Ti, Mn look rather as “Chi-squared”. The axes of plots must be named.
#22 pictures in Figure 10 are of poor quality
#23 line 294 – do not begin the sentence with a number
#24 lines 323-325 and Table 6 – the accuracy should be in the level of 0.001%, because this accuracy is in Table 6; in Table 6 for Cr should be 0.200 (in “deformation” line) – accuracy at level 0.001%; add the next line to the table with the values of differences and some comment under the table concerning the results
Author Response
The main advantage of this article is its completeness and explanation of all aspects, starting with the development and validation of a measurement method, ending with its practical application. I have no major objections to the content of this paper. However, in order to improve its quality and avoid misunderstandings, I propose a number of changes, which I have included in the comments below.
Overall remarks:
- Editorially, the whole text needs to be corrected in some parts in detail, e.g. voltage should be in “kV”, not “kv”; there are sometimes wrong punctuation marks; RSD sometimes is written by capital letters and in other places by lowercase “rms”; and others. Moreover, wrong Tense was used in some places e.g. line 172.
The details that need to be corrected have been modified.
All “kv” has been modified to “KV” in the article.
The “RSD” has been modified to “rsd” in the figure.
The incorrect tenses have been corrected, the present simple has been changed to the past tense in line 172
- Some sentences are very long (4 lines of text) and it cause, that finally are difficult to understand the main meaning of the sentence
Some long sentences in the article have been modified, so that they are easier to understand
Detailed remarks:
#1 line 20, 128 – correct “kv” into “kV”
The “kv” has been modified to “KV” in line 20, 128
#2 line 21 – phrase “the content deviation of all elements” is difficult to understand. If it is a difference between the sum of all measured contents of the elements and the 100% or if the summary of all individual deviations? It must be clearly described. (see also remark #17)
“the content deviation of all elements” is a difference of measured content of elements from standard value. In order to make it easy to understand, “Content deviation” has been changed for “the sum of differences between measured values and standard values for all elements using calibration coefficient is only 0.107%”.
#3 lines 25-28 – all percentages should be in the same level of accuracy 0.01% or 0.001% (see also remark #24)
The accuracy was determined to be 0.001%.
#4 lines 29-30 – delete “Optimization parameter” and “Calibration coefficient” – they are too general;
These general key words have been deleted.
#5 lines 36-37 – correct the phrase “on the same plane and different planes along the thickness direction” to more understandable
This phrase has been deleted.
#6 line 44 – what is it “plastic shape” ?
The “plastic shape” has been changed to “plasticity”.
#7 lines 48-50 – the sentence “Common analysis methods for component distribution, such as scanning electron microscope, energy spectrum analysis and electron probe microanalysis, are only aimed at micro area test, with slow analysis speed and low quantitative sensitivity” is not true. Firstly, SEM is not the method of chemical analyse (but electron beam of SEM is used as a source of high energy electrons). What is mean “slow speed” – if 20 s is slow or fast? What is mean “low quantitative sensitivity” and if 0,1 % is low? These comments are too vague and can be compared to another method rather than as imperative statements.
These vague comments have been modified as follow: Common analysis methods for component distribution, such as energy dispersive spectroscopy and electron probe microanalysis, are aimed at precise test and are used for single points and small regions characterization.
#8 line 81 – data placed in table were measured, from standard or from otherwise? This information must be provided. (reference)
The data placed in table were measured measured by GB/T 20975-2020.The explanation has been added to the article.
#9 line 88 – what does mean “target material was Rh”? Especially, what does mean “Rh”
It is a part of instrument that absorbs incoming electrons, and the“Rh” means the element rhodium in the periodic table.
#10 line 90 – the pressure in chamber should be in SI unit [Pa].
The “20 mbar” has been changed to “2000pa”.
#11 lines 111-113 – there is not explained, what does mean “positive” and “negative” segregation.
Changes have been made in the manuscript as follow (lines 126-127): when the confidence interval is 97.5%, DS is positive segregation, when the confidence interval is 2.5%, DS is negative segregation. A clear description has been added to the article.
#12 line 122 – why X was selected as 0.15?
According to GB/T37793-2019, the cumulative frequency X can be 0.05~0.2, if there are no special provisions, X is recommended to be 0.15 (lines 137-138).
#13 Table 2 – lack of units; not “Electricity” but “Current”
The missing units have been added in Table 3. “Electricity” has been changed to “Current” in this manuscript.
#14 line 128 – the current and voltage cannot be “0” – in table 2 are given minimal values
The minimum voltage has been changed to 10kV and the minimum current has been changed to 100μA.
#15 figure 3 - the horizontal axis and vertical “blue” are not described
The horizontal represents the group number for the orthogonal experiment, and vertical “blue” represents rcd. Figure 3 has been modified.
#16 figures 4 and 5 should be modified – at the horizontal axis are the values of current; lack of the Al signal, described in text
The horizontal axis of Figures 4 and 5 have been added.
Description about Al is explained as follows: due to the high content of matrix elements and strong peak-to-back ratio signal, Al do not belong to the key elements to be investigated. Therefore, the sum of the peak-to-back ratios shown in Figure 4 does not include the matrix.
#17 lines 207-220 – there must be clearly defined if the % deviation is the deviation of the slope or it is deviation of the measured content of elements from standard value. Both types of deviations are used in text, but e.g. for (Zr, Ti) the deviation concerns rather slope than “measured value” -> see next sentence, where the deviation concerns the elements. You must avoid misunderstandings, like this. Moreover, there must be placed information in which way the value of “the content deviation of all elements” (0.107%) was determined (see. remark #2)
In order to avoid misunderstanding, we have changed the expression “deviation of the measured content of elements from standard value” to “Content difference”.
The value of “the content deviation of all elements” (0.107%) was determined by a “Content difference” row in Table 4.
#18 table 3 – there is lack of Mg. Moreover, the sum of elements, both certified (!) and measured is not 100% (app. 97,6%). Why? Place in table additional line with percentage deviation, because you comment it in line 220.
Because there are other elements in the E2323C standard sample, the eight elements selected in Table 4 are the elements existed in 7B05.
A new“Content difference”row has been added in Table 4
#19 lines 226-227 – there should be the reference to Fig. 1 and the directions marked over there (ND, RD, TD)
The directions marked has been supplemented in Figure 7
#20 concerns the data placed in Table 4 - How were determined the values of S, DS positive, DS negative and SRx, referring to eq. (1), (2) and (3). In the Table 4 are not the symbols like used in the equations. There must be explanation or have to be written some additional formulas.
Reference formulas 1, 2 and 3 have been added to determine the data in Table 5: the values of S and SRx refer to eq. (1) and (3). The values of DS Positive and DS Negative correspond to confidence intervals of 97.5% and 2.5% respectively, referring to eq. (2). The results are shown in Table 5.
#21 lines 257-258 and figure 8 – it is rather the “density distribution” or “count distribution” than “frequency distribution”; please check it, because “frequency” is connected rather with wavy phenomena. The distributions model of Fe, Ti, Mn look rather as “Chi-squared”. The axes of plots must be named.
The “frequency distribution” In the whole article has been replaced with “count distribution”, and the horizontal axis in Figure 8 has been marked as “Concentration (ω/%)” ,and the ordinate has been marked as “Counts”.
#22 pictures in Figure 10 are of poor quality
A clear image has been uploaded in this manuscript.
#23 line 294 – do not begin the sentence with a number
The phrase of “in this paper” has been added at the beginning of the sentence.
#24 lines 323-325 and Table 6 – the accuracy should be in the level of 0.001%, because this accuracy is in Table 6; in Table 6 for Cr should be 0.200 (in “deformation” line) – accuracy at level 0.001%; add the next line to the table with the values of differences and some comment under the table concerning the results
The accuracy has been modified in the level of 0.001%, and the information about content differences has been added in Table 7.
Round 2
Reviewer 1 Report
Thanks to the authors for the answers to the proposed comments. Although almost all them have been considered, I think that the most important point (comment number 14) should be adressed before consider the manuscript for publication. This kind of analysis with a direct comparison with the data obtained by the proposed model in the manuscript would improve exponentially the quality of this work.
Author Response
- It would be interesting to check the quantities obtained using the developed method with common analysis methods for component distribution, such as EDS.
1.EDS was used to check the quantities obtained using the XRF method. The results were presented in the last section of the paper.
2.The language of the article has been polished.
3.Some results were not multiplied by calibration factors(μ), so content data were also modified in this version.
